# A Plant Biostimulant from *Ascophyllum nodosum* Potentiates Plant Growth Promotion and Stress Protection Activity of *Pseudomonas protegens* CHA0

**DOI:** 10.3390/plants12061208

**Published:** 2023-03-07

**Authors:** Jai Singh Patel, Vinodkumar Selvaraj, Prashant More, Ramin Bahmani, Tudor Borza, Balakrishnan Prithiviraj

**Affiliations:** Department of Plant Food and Environmental Sciences, Dalhousie University, Halifax, NS B3H 4R2, Canada

**Keywords:** salinity tolerance, growth promotion, *Ascophyllum nodosum*, *Pseudomonas protegens* CHA0, fucoidan, root colonization

## Abstract

Abiotic stresses, including salinity stress, affect numerous crops, causing yield reduction, and, as a result, important economic losses. Extracts from the brown alga *Ascophyllum nodosum* (ANE), and compounds secreted by the *Pseudomonas protegens* strain, CHA0, can mitigate these effects by inducing tolerance against salt stress. However, the influence of ANE on *P. protegens* CHA0 secretion, and the combined effects of these two biostimulants on plant growth, are not known. Fucoidan, alginate, and mannitol are abundant components of brown algae and of ANE. Reported here are the effects of a commercial formulation of ANE, fucoidan, alginate, and mannitol, on pea (*Pisum sativum*), and on the plant growth-promoting activity of *P. protegens* CHA0. In most situations, ANE and fucoidan increased indole-3-acetic acid (IAA) and siderophore production, phosphate solubilization, and hydrogen cyanide (HCN) production by *P. protegens* CHA0. Colonization of pea roots by *P. protegens* CHA0 was found to be increased mostly by ANE and fucoidan in normal conditions and under salt stress. Applications of *P. protegens* CHA0 combined with ANE, or with fucoidan, alginate, and mannitol, generally augmented root and shoot growth in normal and salinity stress conditions. Real-time quantitative PCR analyses of *P. protegens* revealed that, in many instances, ANE and fucoidan enhanced the expression of several genes involved in chemotaxis (*cheW* and *WspR*), pyoverdine production (*pvdS*), and HCN production (*hcnA*), but gene expression patterns overlapped only occasionally those of growth-promoting parameters. Overall, the increased colonization and the enhanced activities of *P. protegens* CHA0 in the presence of ANE and its components mitigated salinity stress in pea. Among treatments, ANE and fucoidan were found responsible for most of the increased activities of *P. protegens* CHA0 and the improved plant growth.

## 1. Introduction

Crop plants can be challenged by various abiotic stress factors, such as salinity, drought, waterlogging, and soil pH. Among these factors, salinity represents a major threat, causing a reduction in the productivity of many crops. Salinity affects almost half of the irrigated areas, severely impacting many important crops that are sensitive to low salinity levels, drastically limiting their yield [1,2,3,4]. Under salinity stress, the accumulation of Na^+^ and Cl^−^ creates an imbalance of nutrients, affecting a wide range of essential physiological activities [5].

Pea (*Pisum sativum*) is one of the essential leguminous crops in developing countries, due to its high content of proteins, starch, and dietary fibers [6]. Most cultivated pea varieties are sensitive or moderately susceptible to salinity stress. Irrigation with water containing high salt concentrations results in reduced plant growth and an increased root-to-shoot ratio [7]. During the growth season, variations in the concentration of salt in the soil can significantly harm the plants. Several other factors, such as waterlogging, can increase the damage created by salinity stress [8]. Typical symptoms of salinity stress include leaf yellowing, necrosis in older leaves, reduction in germination percentage, early death of the young seedling, decrease in nodulation, and ultrastructural modifications of chloroplasts and mitochondria, as well as changes in photosynthesis and antioxidative metabolism [7,8].

Plant biostimulants, including natural compounds, such as protein hydrolysates, humic and fulvic acids, chitosan, and extracts obtained from seaweed, along with rhizospheric microorganisms that comprise bacteria and fungi have been shown to alleviate the detrimental effects of salinity stress [9,10,11]. Extracts obtained from seaweed, especially from brown macroalgae (Phaeophyceae), are increasingly studied and used for their biostimulant activity [10,11,12,13,14,15,16]. Seaweed extracts can stimulate both the primary and secondary metabolic processes of plants, leading to improved nutrient uptake and assimilation under a variety of stresses [9,14,17]. Several studies indicated that brown macroalgae *Ascophyllum nodosum* extracts (ANE) can promote the uptake of micronutrients and macronutrients and the accumulation of phytochemicals, such as anthocyanins and phenolics, which are likely to be responsible for the enhancement of plant tolerance against different stressors [9,14,17,18,19]. ANE was shown to affect proline metabolism in common bean plants (*Phaseolus vulgaris*) and, therefore, to increase the tolerance of plants to drought stress [20]. Soybean (*Glycine max*) treated with ANE demonstrated improved tolerance to drought stress, including higher relative water content, stomatal conductance, and antioxidant activity; ANE also changed the expression of several stress-responsive genes [16]. Seaweed extracts can induce different plant defense pathways, including the activation of the jasmonic acid-dependent signaling pathway [14,15,21,22]. ANE was also found to mitigate salinity stress in the model organism Arabidopsis [13,14,23], as well as in several crop plants, such as tomato [24], passion fruit [25], and avocado [26]. Seaweed extracts were found to improve the survival of Kentucky bluegrass (*Poa pratensis* cv. Plush) [27] and of chickpea (*Cicer arietinum*) [28], due to the accumulation of amino acids, including proline, and enhanced activities of enzymes involved in limiting the effects of oxidative damage [28]. Fucoidan and alginate are major complex polysaccharides found in brown seaweed [19,29,30,31,32,33]. Mannitol, a sugar alcohol, is also highly abundant in these macroalgae, acting as an osmolyte and antioxidant [31,32,33,34]. Though complex brown macroalgae extracts, such as ANE, were found to elicit various mechanisms in plants which can alleviate biotic and abiotic stresses, the potential biostimulant effect of brown seaweed components, such as fucoidan, alginate, and mannitol, have not been sufficiently explored [14,19,29].

Production of phytohormones and of other bioactive molecules, as well as enhanced colonization of the rhizosphere by beneficial soil bacteria, and of plants by endophytic bacteria, can increase plant growth under salinity stress [35,36]. For example, *Pseudomonas putida* strain S1 and *Pseudomonas aeruginosa* strain Crg have been shown to mitigate salinity stress in chickpea [37], and *Pseudomonas* PS01 in Arabidopsis [38]. Also, *Pseudomonas* sp. possessing 1-aminocyclopropane-1-carboxylate [39] deaminase activity improved tolerance and increased yield under salinity stress in tomato plants [40] and enhanced salt tolerance in *Camelina sativa* [41]. Seed biopriming with *Aneurinibacillus aneurinilyticus* and *Paenibacillus* sp. was found to alleviate salinity stress in French bean [42]. Not only rhizospheric, but also endophytic microorganisms, such as *Arthrobacter* sp., *P. putida* and *Bacillus* sp., can increase salt tolerance in plants (reviewed by [35]).

In the future, several alternative stress mitigation technologies are likely to play a significant role in improving crop production [9,14,19,35]. Several studies pointed out that abiotic stresses, such as salinity stress, can be reduced by using seaweed extracts and various beneficial bacteria. Less is known, however, about the effects of seaweed extracts on rhizosphere bacteria and whether these extracts can influence the interaction between plants and beneficial microbes. The aim of this study was to determine the effects of ANE, fucoidan, alginate, and mannitol on *Pseudomonas protegens* CHA0 secretion and colonization ability and the combined effects of these seaweed compounds and *P. protegens* CHA0 on the growth of pea (*P. sativum*). The effects were tested in the absence and presence of salt stress. The inclusion of fucoidan, alginate, and mannitol in the experiments was done to determine whether, individually, these major components of ANE can have effects similar to those of ANE. Biochemical and gene expression analyses of *P. protegens* CHA0 and plant growth experiments indicated that ANE and fucoidan have the most pronounced beneficial effects.

## 2. Results

### 2.1. Screening of Bacterial Strains for Different Plant Growth-Related Biological Activities

Seventeen bacteria were screened for their activities by plate assays. Among them, *P. protegens* CHA0 and Sinorhizobium meliloti RM11559 were excellent performers in all activities, including phosphate solubilization, siderophore production, hydrogen cyanide (HCN) production, and indole-3-acetic acid (IAA) production. However, overall, *P. protegens* CHA0 was better in all the assays compared to S. meliloti RM11559 and the other bacterial cultures (Table 1). As a result, *P. protegens* CHA0 was selected for further studies.

### 2.2. Effect of ANE, Fucoidan, Alginate, and Mannitol on Siderophore Production, Phosphate Solubilization, HCN Production, and IAA Production by P. protegens CHA0

*P. protegens* CHA0 was assessed using qualitative and quantitative assays for different activities under salinity stress.

Siderophore activity was determined first by the halo zone formed around the culture inoculums. Concentrations of 0.1% ANE and 0.01% fucoidan, alginate, and mannitol were found to form halo zones of comparable sizes (Figure 1A). However, quantification of siderophore production in the presence of 0.1% ANE and 0.01% fucoidan, alginate, and mannitol showed that the highest amounts were produced in the presence of ANE (57.66 psu), followed by fucoidan, mannitol, and alginate; siderophore amounts were significantly higher in all compounds as compared to control (Figure 1B).

Similar to siderophore production, in the plate assay, the addition of 0.1% ANE and 0.01% fucoidan, alginate, and mannitol was clearly associated with substantial phosphate solubilization (Figure 2A). Quantitatively, fucoidan, ANE, and mannitol were found to be better than the control, while alginate showed no significant change in phosphate solubilization (Figure 2A,B).

HCN production is related to defense mechanisms employed by *P. protegens* CHA0. HCN production was high in the presence of mannitol, alginate, and fucoidan as compared to ANE and control (Figure 3A,B).

The IAA production assay provides information on the growth promotion activity of *P. protegens* CHA0. The effect of media amendments was very similar to that observed in phosphate solubilization and siderophore production; that is, ANE (9.6 µg/mL) and fucoidan (9.37 µg/mL) were found to enhance the most IAA production. Alginate was slightly better than the control, while mannitol did not improve IAA production (Figure 4).

### 2.3. Effect of ANE, Fucoidan, Alginate, and Mannitol on CFU Count of P. protegens CHA0

The CFU count of *P. protegens* CHA0 was found to be increased by the addition of *A. nodosum* components, in the presence and absence of salt stress (Figure 5A,B). The count was found to be increased more than two-fold by 30 mM NaCl. Surprisingly, in salinity conditions, mannitol failed to enhance bacterial growth, though it did so significantly in normal conditions. ANE and fucoidan showed the most consistent effects in both growth conditions (Figure 5B).

### 2.4. Effect of ANE, Fucoidan, Alginate, and Mannitol, in the Absence and Presence of Salinity Stress, on Key Genes of P. protegens CHA0 Involved in Plant Growth Promoting Rhizobacteria (PGPR) Activity

The gene *cheW* codes a protein involved in chemotaxis that is present in a wide range of bacteria [43]. In absence of salt, all treatments induced the gene expression level compared to the control; however, gene up-regulation was significant only in the alginate and mannitol treatments (7.15 fold and 4.27 fold, respectively; Figure 6A). In presence of salt stress, alginate, and ANE treatments showed the highest expression (4.20 fold and 4.74 fold, respectively) compared to the control, while fucoidan and mannitol treatments had almost no effect on gene expression (Figure 6A).

WspR, the product of the *WspR* gene, is the response regulator of the Wsp chemosensory system in several bacteria [44,45]. The expression of *WspR*, in absence of salt stress, was found to be significantly increased in mannitol, alginate, and fucoidan treatments (3.17 fold, 3.06 fold, and 2.56 fold, respectively), but much less in ANE treatment. In the presence of salt stress, the expression pattern changed noticeably, with fucoidan treatment having the strongest up-regulatory effect (3.79-fold increase), while ANE, alginate, and mannitol treatments showed only minimal effects. (Figure 6B).

The gene *hcnaA* codes a hydrogen cyanide synthase that is responsible for the volatile HCN production by various bacteria [46,47,48]. HCN is an antimicrobial compound that is also released by *P. protegens* CHA0. The highest expression of *hcnaA*, in absence of salt stress, was observed in the ANE treatment (14.4-fold), followed by the treatments with fucoidan and alginate. Mannitol treatment was not different from the control. In the presence of salt stress, compared to the control, the expression of *hcnaA* was induced by ANE and alginate treatments (5.72-fold and 5.48-fold, respectively), but much less by fucoidan (2.93-fold). Mannitol treatment was not different from control in this condition as well. Overall, versus control, the expression of *hcnaA* was less pronounced in all the treatments under salinity conditions, except for the treatment with alginate (Figure 6C).

The gene *pvdS* is responsible for the production of sigma factor PvdS, which is involved in the positive regulation of secondary metabolite biosynthetic processes, including pyoverdine synthesis [49,50]. In the absence of salt stress, an overall increase in *pvdS* transcript abundance was observed in treated plants, but these differences were not significant when compared to the control. In the presence of salt stress, the highest expression was observed in mannitol treatment (2.10-fold), but this was the only significant difference compared to the control, among treatments (Figure 6D).

### 2.5. Effect of P. protegens CHA0 on Different Plant Growth-Related Parameters under Salt Stress

The effects of ANE, fucoidan, alginate, and mannitol on plant growth, in the absence and presence of *P. protegens* CHA0, and salinity stress, were assessed by growing the pea plants in a hydroponic system (Figure 7). After 7 days in the hydroponic system, the fresh weight of roots and shoots was recorded (Figure 8A,B). All of the treatments and the control showed significantly higher root fresh weight (FW) in the presence of *P. protegens* CHA0 in the normal and salinity stress conditions (Figure 8A). Among treatments, ANE and fucoidan were better in all conditions (presence or absence of *P. protegens* CHA0 or/and salinity stress) as compared to alginate and mannitol. In the absence of salt stress and of *P. protegens* CHA0, only alginate was not different from the control while, in the presence of *P. protegens* CHA0, alginate and mannitol were not statistically different from the control (Figure 8A).

While the presence of *P. protegens* CHA0 had a significant effect on root FW, the overall effects on shoot FW were not that obvious (Figure 8A,B). Overall, fucoidan, ANE, and mannitol improved shoot growth, while alginate was found to be better than the control, only in the presence of *P. protegens* CHA0 and salinity. Also, ANE and fucoidan performed better in the presence of *P. protegens* CHA0, in both the absence and presence of salt stress (Figure 8B).

## 3. Discussion

Tritrophic interaction between root-soil and rhizosphere microorganisms is a crucial phenomenon affecting plant growth, health, and defense. Several physiological activities of plants, such as nutrient uptake, can be improved by rhizosphere microbes, which can reduce the use of agrochemicals [51]. Microorganisms are recruited by exudates produced by root cells. Root exudates are a rich source of fixed carbon, such as polysaccharides, and of chemicals related to chemotaxis for the microorganisms found in the rhizosphere. Rhizospheric microorganisms perceive the signals produced by the plant roots and produce a wide range of chemicals in response to these signals, promoting plant growth and enhancing their tolerance against biotic and abiotic stresses [52,53,54].

The results of the current study indicate that ANE, and components present in ANE (fucoidan, alginate, and mannitol), can (i) enhance the production by *P. protegens* CHA0 of essential products associated with root colonization, (ii) promote the colonization of pea root by *P. protegens* CHA0, and (iii) mitigate salt stress in pea, a process that is further improved in the presence of *P. protegens* CHA0. This last effect was found to be more pronounced in root growth promotion than shoot growth. *A. nodosum* extract (ANE) has been shown to exhibit strong biostimulant activity against several biotic and abiotic stresses [13,14,19,23,55]. However, the effects of fucoidan, alginate, and mannitol, which are main components of *A. nodosum*, and therefore of ANE, are less characterized, though oligosaccharides and polysaccharides present in ANE have been shown to be elicitors of plant defense responses [56]. The role of ANE in the potentiation of beneficial bacteria growth and root colonization is still unexplored. On the other hand, several species from the *Pseudomonas* genus are well known to enhance plant defense mechanisms against abiotic and biotic stresses [37,57,58].

The presence of ANE improved the PGPR activities of *P. protegens* CHA0. Siderophores are well-known high-affinity iron-chelating compounds, which facilitate iron uptake from soil by plants. Plants assimilate iron from the soil through the roots [59,60,61]; siderophore production by *P. protegens* CHA0 can enhance iron transport across the plasma membrane of root epidermal cells and, therefore, the assimilation process. Results from the current study showed that all treatments improved siderophore production of *P. protegens* CHA0, though, overall, ANE was found to perform the best. It is likely that the more complex composition of ANE can explain this effect, suggesting potential synergistic effects of the components present in ANE. Pyoverdines are major fluorescent siderophores in *Pseudomonas* spp. [61,62], and *pvdS* is a key gene in the regulation of pyoverdine synthesis [49,50]. All treatments, including that with ANE, had a rather limited effect on the expression of *pvdS*, and the addition of salt in the medium determined changes in its abundance. The lack of a positive correlation between siderophore production by *P. protegens* CHA0 under different treatments and *pvdS* expression suggests that several other genes related to pyoverdines biosynthesis, secretion, and iron uptake [50,63] may contribute to the observed phenotype; also, inconsistencies can occur when comparing and integrating transcriptomics, proteomics, and metabolomics data [64,65,66].

Phosphorus (P) is a major limiting factor for plant growth. Insufficient amounts of phosphorus in agricultural soils are often counterbalanced by the excessive use of high-priced P-based chemical fertilizers [67]. Organic acids produced by phosphate-solubilizing bacteria can make the P present in the soil more accessible to plants [68,69]. Among the *A. nodosum* components analyzed, the phosphate solubilizing activity of *P. protegens* CHA0 was found to be enhanced in the presence of fucoidan and ANE, suggesting that fucoidan is the main component in ANE responsible for potentiating this activity in *P. protegens* CHA0.

Beneficial *Pseudomonas* spp. have been reported to suppress different plant diseases by producing antimicrobial metabolites, such as hydrogen cyanide [48,69]. The production of HCN by *P. protegens* CHA0 was not influenced by ANE, though mannitol, alginate, and fucoidan were found to increase around 3-fold this activity. It is likely that ANE components, other than polyols and complex polysaccharides, counterbalanced the enhancing effects determined by mannitol, alginate, and fucoidan. Interestingly, in absence of salt stress, the expression of *hcnaA* was found to be the highest in presence of ANE, a pattern that is opposite to that of HCN production. The *hcnaA* encodes subunit A of hydrogen cyanide synthase, responsible for the volatile HCN production by bacteria. The functional protein is a heterotrimer, formed by subunits HcnA, HcnB, and HcnC [46,47,48]. It is difficult to rationalize the effects triggered by ANE, which had no effect on HCN production, but enhanced the expression of *hcnaA*. One possible explanation is that ANE compounds, such as polyphenols, had inhibitory effects on the enzymatic activity of HCN synthase.

Plant growth promotion is one of the important effects of beneficial PGPR. Production of phytohormones, such as IAA, is the key mechanism behind growth promotion by PGPR. Several studies provided support for IAA production by PGPR [70,71,72]. IAA production ability of *P. protegens* CHA0 was found to be enhanced in the presence of ANE; fucoidan and alginate, but not mannitol, were found to have similar effects. Similarly to the results observed in siderophore production, the better activity of ANE might be explained by the potential synergistic effects of fucoidan and alginate.

Plants are essential sources of nutrition and provide shelter for rhizospheric bacteria. Several factors affect the colonization of microorganisms in the rhizosphere of plants. Rhizobacteria can be attracted and move toward plant roots by several chemoattractants secreted by plants [51,73,74]. Chemotaxis allows bacteria to move along the signal gradient [75], and the final number of cells colonizing the root can be in the range of 10^11^ cells per gram of root [76]. In absence of salt stress, pea root colonization by *P. protegens* CHA0 was not strongly enhanced, though mannitol, ANE, and fucoidan had some stimulating effects. In the presence of salt stress, root colonization was promoted by ANE and fucoidan, while the effects of mannitol and alginate were limited. These results indicate that (i) the addition of salt influences the effects of *A. nodosum* components, and (ii) *A. nodosum* components influence differently the various *P. protegens* CHA0 functions related to root colonization. Gene expression analysis of *cheW* and *WspR* in *P. protegens* CHA0, after treatment with ANE and components, provided limited support to pea root colonization; overall, no clear positive correlation between transcript abundance and enhanced physiological responses could be determined. CheW is one of the proteins involved in the transmission of sensory signals from the chemoreceptors to the motor proteins of the flagellar apparatus, which is responsible for the movement of bacteria toward the rhizosphere [43]. The WspR chemosensory system is well characterized in bacteria; WspR is the response regulator of this system. WspR contains the conserved GGDEF domain, which is involved in the formation of the intracellular-signaling molecule cyclic diguanylate (c-diGMP) [44,45]. Increased levels of c-diGMP result in enhanced biofilm formation and, therefore, of root colonization [44]. ANE-enhanced root colonization in the absence and presence of salt stress; however, treatments with ANE determined a large increase of transcript abundance only in the case of *cheW*, in the presence of salt stress. Fucoidan also had significant effects on root colonization, but determined the up-regulation of *WspR*. It is likely that, to have a more comprehensive view, at a molecular level, of the effects of ANE and its components on root colonization, more genes have to be analyzed, or different omics approaches should be considered because of the rather complicated relationship between transcriptomics, proteomics, metabolomics, and phenotype data [64,65,66].

The effects of the treatments with ANE, fucoidan, alginate, and mannitol were found to be more pronounced on root growth compared to shoot growth, and these effects on root growth were further enhanced in the presence of salt stress. The addition of *P. protegens* CHA0 also had a marked effect on root growth, while the overall effects on shoot growth were more limited. ANE is well known for its biostimulant activity [29,77,78,79,80], including improved seed germination, seedling vigor, and plant growth [14,19]. Mannitol is naturally synthesized in numerous bacteria, fungi, algae, and land plants, and functions as an osmolyte, energy storage, and antioxidant [34,81,82]. Mannitol treatment can improve salt tolerance in different plant species [83,84]. However, in the current work, except for shoot-fresh weight under the normal condition, which was marginally enhanced, mannitol treatment did not significantly influence root or shoot growth under both normal and stress conditions. Slightly better results have been observed on shoot growth in the presence of *P. protegens* CHA0. These results can be ascribed to the low concentration of the mannitol used in this study, that is, 0.01% mannitol; this concentration is equivalent to ~0.5 mM mannitol, which is quite low compared to the concentration of mannitol proved to act as an osmolyte [83,84]. The present study supports these findings, as ANE and fucoidan consistently improved root and shoot growth in the absence and presence of salinity stress, while *P. protegens* CHA0 further augmented these effects.

## 4. Materials and Methods

### 4.1. Plant Material and Growth Conditions

Seeds of pea (*Pisum sativum* L. cv. Sabre) were used in the experiments. Pea seeds, selected to have a weight of around 0.35 g, were surface-sterilized with 1% of sodium hypochlorite for 3 min, followed by rinsing in a series of distilled water. Seeds were soaked in distilled water overnight. Imbibed seeds were incubated in moist chambers for germination under a 16/8 light and dark period at 22 ± 2 °C. Germinated seeds with a radicle length of 2–4 cm were transferred to the hydroponics system containing ¼ MS media (without sucrose; pH 5.8).

### 4.2. Source of ANE, Fucoidan, Alginate, and Mannitol

Acadian^®^, a commercial ANE product, was provided by Acadian Seaplants Limited, Dartmouth, NS, Canada. Fucoidan was obtained from Marinova (Marinova Pty Ltd., Cambridge, TAS, Australia). The sodium salts of alginate and mannitol were from Sigma-Aldrich. (Oakville, ON, Canada).

### 4.3. Bacterial Cultures Screening

Seventeen beneficial bacterial cultures (Table 1) were characterized for their various biochemical activities, such as phosphate solubilization, siderophore production, IAA production, and HCN production. Bacterial cultures were provided by Dr. Zhenyu Cheng, Dalhousie University, NS, Canada. All the bacterial cultures were grown on their respective media (Appendix A) at 28 ± 2 °C.

### 4.4. Siderophore Production Assay

Siderophore production assay was performed by using the Chrome Azurol S dye (CAS, Sigma-Aldrich) method described by Schwyn and Neilands [85]. This assay estimates siderophore production qualitatively and quantitatively. The qualitative test was done on King’s B agar media containing the CAS solution at a ratio of 1:15. Aqueous suspensions of bacterial cultures with a CFU count of 2.4 × 10^8^ were spotted on the middle of CAS agar plates in triplicate. All Petri dishes were incubated at 28 ± 2 °C temperature for 48 h. The formation of an orange halo zone around the developing bacterial colonies indicated a positive reaction for the assay. For quantitative estimation, 0.5 mL of the culture supernatant was added to 0.5 mL of CAS solution, followed by 10 µL of sulfosalicylic acid (0.2 M) (shuttling reagent). After 30 min of incubation at room temperature, the optical density was recorded at 630 nm. Siderophore production was quantified in terms of percent siderophore units (psu), using the formula: (Ar − As)/Ar × 100; where Ar is the absorbance of the reference solution, and As is the absorbance of samples.

### 4.5. Phosphate Solubilization Assay

Phosphate solubilization by microbes was assayed using the NBRIP-BPB medium described by Nautiyal [86]. Bacterial strains with CFU count of 2.4 × 10^8^ were inoculated on solid NBRIP-BPB medium in triplicate and incubated for 2–3 days at 28 ± 2 °C. The clear zone around the growing colonies was considered as a positive result for phosphate solubilization.

Quantitative estimation of phosphate solubilization was carried out in broth medium NBRIP, amended with different treatments in triplicates. The amount of released phosphate by the activity of *P. protegens* CHA0 was estimated using the method described by Fiske and Subbarow [87].

### 4.6. HCN Production Assay

HCN production assay was performed as described by Lorck [88]. Bacterial strains with a CFU count of 2.4 × 10^8^ were inoculated on the middle of the Petri plate in triplicate. Picrate filter paper (0.5% picric acid in 0.2% *w*/*v* Na_2_CO_3_) was placed on the lid of the Petri plate. HCN formation determines the change of color from deep yellow to orange, and, finally, to dark brown. Bacterial strains changing the color of the picrate filter paper from deep yellow to brown were considered as positive for HCN production. The plates were sealed by parafilm and incubated for 72 h at 28 °C.

Dissolved, free cyanide ions concentration in the liquid medium was determined using the colorimetric methemoglobin method described by von Rohr et al. [89]. The liquid medium and the methemoglobin reagent were mixed at a 1:1 ratio, and the optical density was recorded at 424 nm.

### 4.7. Indole-3-Acetic Acid (IAA) Production Assay

The IAA production assay was performed using the modified protocol described by Malik and Sindhu [90]. Bacterial cultures were grown on their respective media for 48 h and 72 h, respectively, at 28 ± 2 °C. Culture suspension (2.4 × 10^8^ CFU) was centrifuged at 3000× *g* for 30 min, and 2 mL of supernatant was used for the assay. The reaction was performed by the addition of 3 µL of orthophosphoric acid (85%) and 4 mL of Salkowski reagent (50 mL of 35% perchloric acid and 1 mL 0.5 M FeCl_3_). Optical density was recorded at 530 nm.

### 4.8. Protease Assay

Protease assay was performed as previously described by Kembhavi et al. [91]. In brief, the bacterial culture was centrifuged at 5000 rpm for 6 min. The supernatant (150 µL) was transferred to a tube and was mixed with 300 µL of 1% (*w*/*v*) casein (prepared in 20 mM Tris-HCl buffer, pH 7.4) and incubated at 37 °C for 30 min. Then, 0.45 mL of a 10% (*w*/*v*) trichloroacetic acid (TCA) solution was added to stop the proteolysis reaction, followed by incubation of the mixture at room temperature for one hour. Eventually, the reaction mixture was centrifuged at 12,000× *g* for 5 min, and the absorbance of the supernatant was measured at 280 nm. One unit of protease is defined as the amount of enzyme that hydrolyses casein to produce equivalent absorbance to 1 μmol of tyrosine/min with tyrosine as standard [91].

### 4.9. Effect of ANE, Fucoidan, Alginate, and Mannitol on Gene Expression Profiles of P. protegens CHA0

*P. protegens* CHA0 cells were collected after 24 h of growth in the presence and absence of ANE, fucoidan, alginate, or mannitol. RNA isolation was performed using a RNeasy mini kit (Qiagen Inc., Toronto, ON, Canada). Total RNA was quantified with a NanoDrop^™^ 2000 spectrophotometer (Thermo Fisher Scientific Inc., Mississauga, ON, Canada). 2 µg of RNA was used to synthesize cDNA using a RevertAid First Strand cDNA Synthesis Kit (Thermo Fisher Scientific Inc). The relative gene expression of *pvdS* (coding the sigma factor PvdS, which is involved in the regulation of pyoverdine synthesis), *cheW* (CheW protein is implicated in chemotaxis), *wspR* (coding the WspR protein with diguanylate cyclase activity, involved in signal transduction, cell adhesion, and biofilm formation) and *hcnA* (HcnA is an enzyme with glycine dehydrogenase, cyanide-forming activity), was determined by real-time quantitative PCR (RT-qPCR) (QuantStudio 3, Applied Biosystems, Burlington, ON, Canada), using the SYBR^®^ Green Supermix Kit (Bio-Rad Laboratories (Canada) Ltd., Mississauga, ON, Canada). Gene-specific primers were used at a final concentration of 0.1 μM. Primer3 software, version 4.1.0, was used to design the primers used in the experiments (Appendix A). RT-qPCR assays were carried out under the following conditions: denaturation at 95 °C for 2 min, 40 repeats at 95 °C for 20 s, 60 °C for 30 s, and 72 °C for 25 s. Transcript abundance was estimated using the 2^−ΔΔCT^ method [92] for relative quantification by normalizing the data against the endogenous gene (*rpoC*) and the individual with the lowest gene expression in controls. Fold change was calculated with respect to the mean of the CT values of the three biological replicates from treatment C (control). All RT-qPCR experiments were carried out with three biological and three technical replicates.

### 4.10. Effect of ANE, Fucoidan, Alginate, and Mannitol on the Biochemical Activities and Root Colonization Activity of P. protegens CHA0

The concentration of ANE was adjusted to 0.1% [23], while that of fucoidan, alginate, and mannitol were adjusted to 0.01%, considering that the proportion of these compounds in brown seaweed extracts ranges between 2 and 10% [29,30,31,32]. All biochemical assays were performed as mentioned in the previous sections. *P. protegens* CHA0 was grown in the King’s B broth, with incubation at 28 ± 2 °C in a shaking incubator. 

To study the effect of ANE, fucoidan, alginate, and mannitol on the root colonization activity of *P. protegens* CHA0, the CFU count was performed using samples taken from the rhizoplane, after 24 h of stress challenge. 0.2 g of the root sample from each treatment was taken for the experiment. Samples were crushed in 1 mL of sterilized, Millipore water and filtered. 100 µL of 10^−5^ and 10^−6^ dilution was transferred to Petri dishes containing King’s B medium. The plates were incubated at 28 ± 2 °C, and CFU was counted after 24 h.

### 4.11. Effect of ANE, Fucoidan, Alginate, and Mannitol on Plant Growth in the Presence of P. protegens CHA0 under Salinity Stress

Treatments, that is, 0.1% ANE, 0.01% fucoidan, mannitol, alginate, and 25 mL of *P. protegens* CHA0 (2.4 × 10^7^ CFU/mL), were added two days after the germinated seeds were placed in the hydroponic system. Salinity stress was carried out at the same time by the addition of 30 mM NaCl in each treatment. The phenotype of the plants grown in the Magenta jars was observed after eight days of salinity stress challenge. Experiments were repeated 3 times. Each treatment had 5 replicates and there were 4 plants in each replicate.

### 4.12. Statistical Analysis

Analysis of variance (One-Way ANOVA) was performed by using Minitab Statistical Software Version 21.1 (Minitab Inc., State College, PA, USA). A Tukey post hoc test was used to perform multiple mean comparisons at *p* ≤ 0.05.

## 5. Conclusions

The action of ANE and of biological compounds, such as fucoidan, which are found in large amounts in *A. nodosum*, increased the production of antimicrobial compounds and the synthesis of chemotaxis-related chemicals by *P. protegens* CHA0. These results suggest that *A. nodosum* extract, as well as various *A. nodosum* components, can potentiate the beneficial PGP effects of *P. protegens* CHA0 in terms of phosphate solubilization, siderophore, HCN, and IAA production. Colonization of pea roots by *P. protegens* CHA0 was also enhanced, mainly by ANE and fucoidan, under salinity stress. Overall, the combined applications of ANE and *P. protegens* CHA0 greatly improved the growth and tolerance against salinity of pea plants grown in hydroponic conditions. The current study suggests that ANE and compounds such as fucoidan, derived from *A. nodosum*, could be used for the improvement of crop growth under salinity stress. New formulations, containing various formulations of ANE supplemented with beneficial microbes, could also be envisioned for improved yield and plant health.

## Figures and Tables

**Figure 1 plants-12-01208-f001:**
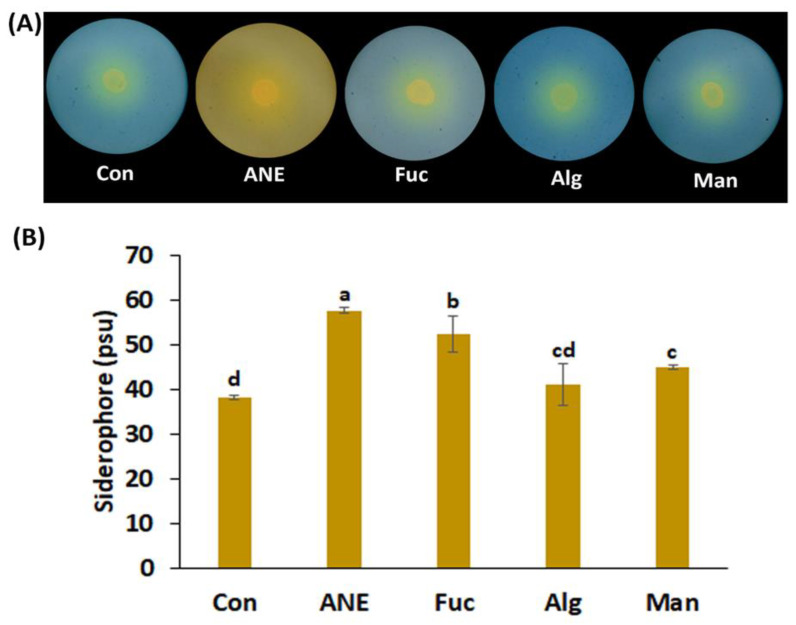
Effect of ANE, fucoidan, alginate, and mannitol on siderophore production by *P. protegens* CHA0. (**A**) Images of the plates showing siderophore production under different treatments. (**B**) Siderophore production quantified using the CAS method. Con, control; ANE, 0.1% Acadian^®^; Fuc, 0.01% fucoidan; Alg, 0.01% alginate; and Man, 0.01% mannitol. Statistical analysis was done using One-Way ANOVA (*p* ≤ 0.05), followed by Tukey’s multiple comparison test (*p* ≤ 0.05). Error bars represent SD. Letter grouping indicates significant differences according to Tukey’s test.

**Figure 2 plants-12-01208-f002:**
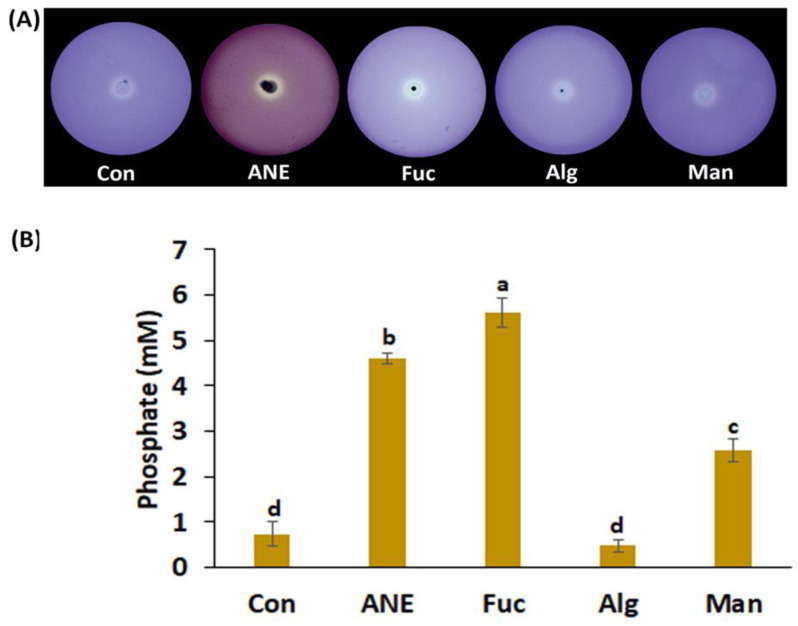
Effect of ANE, fucoidan, alginate, and mannitol on phosphate solubilization by *P. protegens* CHA0. (**A**) Images of the plates showing phosphate solubilization under different treatments. (**B**) Amount of phosphate solubilized by *P. protegens* CHA0 under different treatments. Con, control; ANE, 0.1% Acadian^®^; Fuc, 0.01% fucoidan; Alg, 0.01% alginate; and Man, 0.01% mannitol. Statistical analysis was done using One-Way ANOVA (*p* ≤ 0.05) followed by Tukey’s multiple comparison test (*p* ≤ 0.05). Error bars represent SD. Letter grouping indicates significant differences according to Tukey’s test.

**Figure 3 plants-12-01208-f003:**
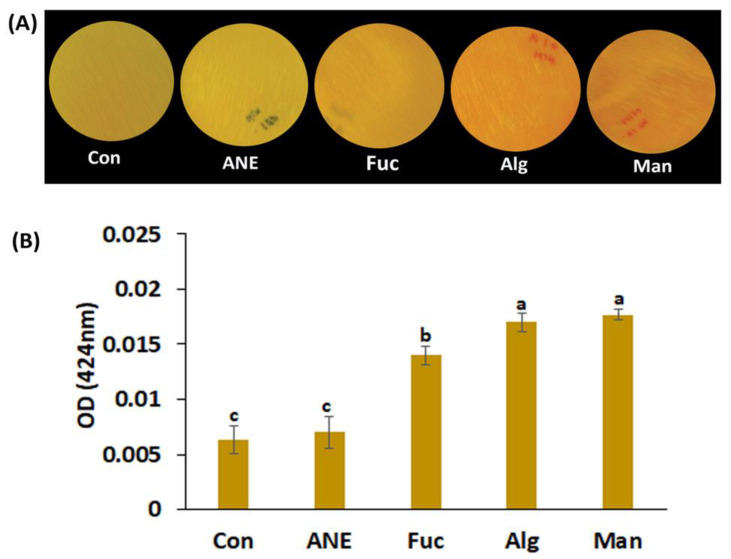
Effect of ANE, fucoidan, alginate, and mannitol on HCN production by *P. protegens* CHA0. (**A**) Images of the plates showing HCN production under different treatments. (**B**) Relative amounts of HCN produced by *P. protegens* CHA0 under different treatments. Con, control; ANE, 0.1% Acadian^®^; Fuc, 0.01% fucoidan; Alg, 0.01% alginate, and Man, 0.01% mannitol. Statistical analysis was done using One-Way ANOVA (*p* ≤ 0.05), followed by Tukey’s multiple comparison test (*p* ≤ 0.05). Error bars represent SD. Letter grouping indicates significant differences according to Tukey’s test.

**Figure 4 plants-12-01208-f004:**
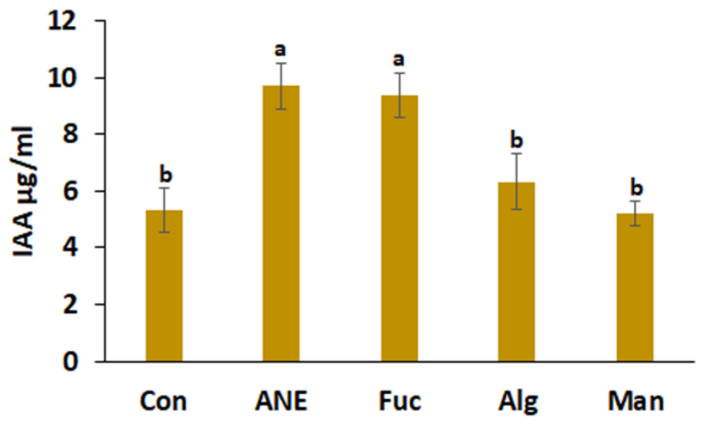
Effect of ANE, fucoidan, alginate, and mannitol on IAA production by *P. protegens* CHA0. Con, control; ANE, 0.1% Acadian^®^; Fuc, 0.01% fucoidan; Alg, 0.01% alginate; and Man, 0.01% mannitol. Statistical analysis was done using One-Way ANOVA (*p* ≤ 0.05), followed by Tukey’s multiple comparison test (*p* ≤ 0.05). Error bars represent SD. Letter grouping indicates significant differences according to Tukey’s test.

**Figure 5 plants-12-01208-f005:**
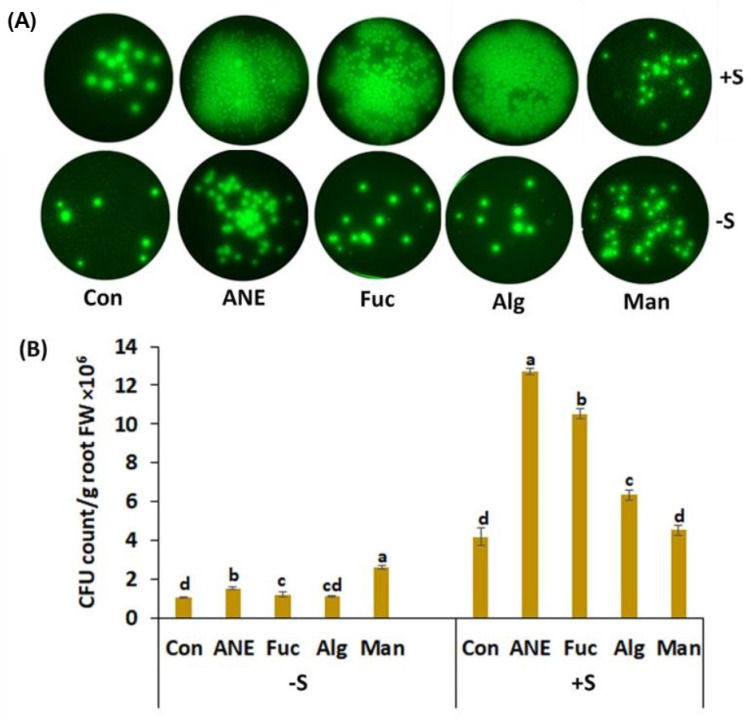
Effect of the ANE, fucoidan, alginate, and mannitol on pea root colonization by *P. protegens* CHA0. (**A**) Images of the plates on which *P. protegens* CHA0 cells were enumerated. (**B**) Colony-forming units (CFU) counts in the absence and presence of salinity stress. Con, control; ANE, 0.1% Acadian^®^; Fuc, 0.01% fucoidan; Alg, 0.01% alginate; and Man, 0.01% mannitol; −S, normal conditions (no salt treatment); +S, 30 mM NaCl. Statistical analysis was done using One-Way ANOVA (*p* ≤ 0.05), followed by Tukey’s multiple comparison test (*p* ≤ 0.05). Error bars represent SD. Letter grouping indicates significant differences according to Tukey’s test.

**Figure 6 plants-12-01208-f006:**
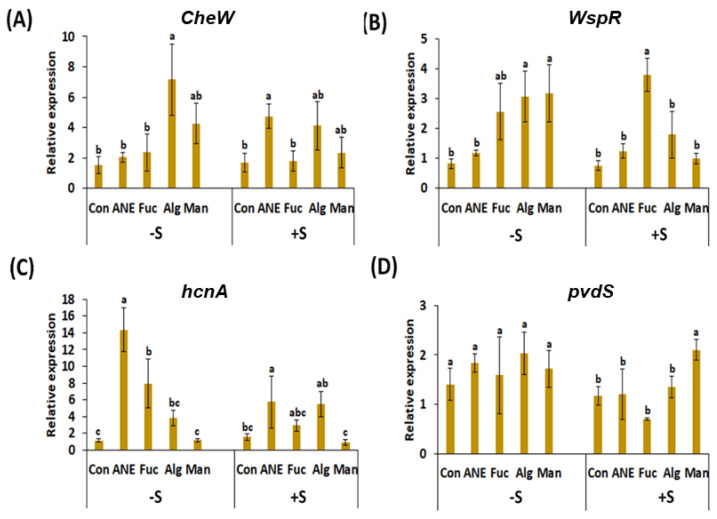
Effect of ANE, fucoidan, alginate, and mannitol on gene expression of *P. protegens* CHA0. (**A**) *CheW*, (**B**) *WspR*, (**C**) *hcnA*, (**D**) *pvdS*. Con, control; ANE, 0.1% Acadian^®^; Fuc, 0.01% fucoidan; Alg, 0.01% alginate; Man, 0.01% mannitol; PP, *P. protegens* CHA0; −S, normal conditions (no salt treatment); +S, 30 mM NaCl. Statistical analysis was done using One-Way ANOVA (*p* ≤ 0.05), followed by Tukey’s multiple comparison test (*p* ≤ 0.05). Error bars represent SD. Letter grouping indicates significant differences according to Tukey’s test.

**Figure 7 plants-12-01208-f007:**
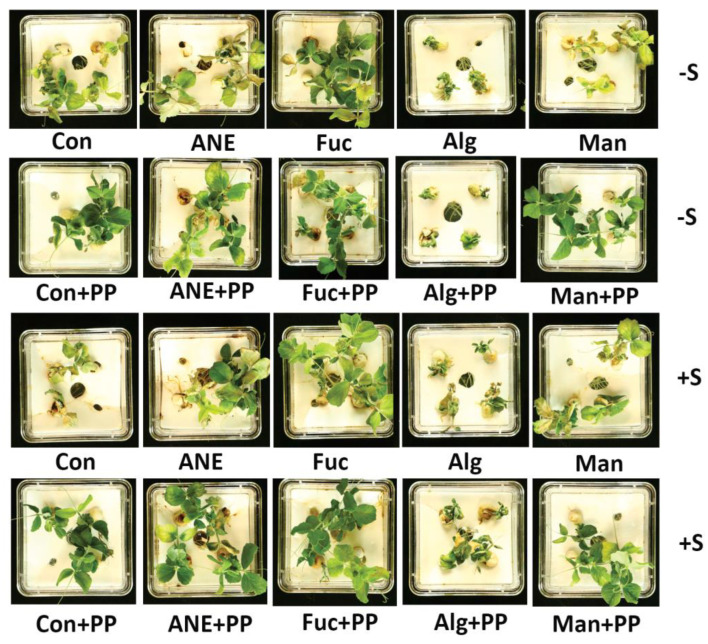
Effect of ANE, fucoidan, alginate, and mannitol on plant growth in the absence and presence of *P. protegens* CHA0 and salinity stress. Con, control; ANE, 0.1% Acadian^®^; Fuc, 0.01% fucoidan; Alg, 0.01% alginate; Man, 0.01% mannitol; PP, *P. protegens* CHA0; −S, normal conditions (no salt treatment); +S, 30 mM NaCl.

**Figure 8 plants-12-01208-f008:**
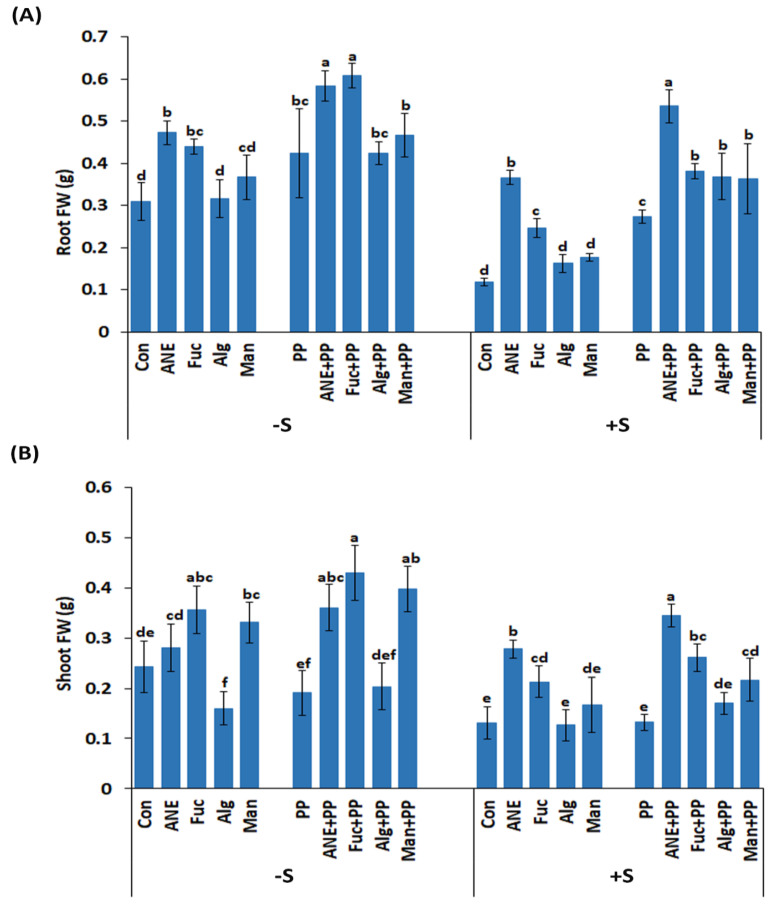
Effect of ANE, fucoidan, alginate, and mannitol on plant growth in the absence and presence of *P. protegens* CHA0 and/or salinity stress. (**A**) Root fresh weight. (**B**) Shoot fresh weight. Con, control; ANE, 0.1% Acadian; Fuc, 0.01% fucoidan; Alg, 0.01% alginate Man, 0.01% mannitol; Pp, *P. protegens* CHA0; −S, normal conditions (no salt treatment); +S, 30 mM NaCl. Statistical analysis was done using One-Way ANOVA (*p* ≤ 0.05), followed by Tukey’s multiple comparison test (*p* ≤ 0.05). Error bars represent SD. Letter grouping indicates significant differences according to Tukey’s test.

**Table 1 plants-12-01208-t001:** Screening of the bacteria for the different biochemical activities. (−) no activity; (+) low activity; (++) medium activity; (+++) strongest activity.

Bacterial Species	Phosphate Solubilization	Siderophore Production	IAA Production	Protease Production	HCN Production
*Azospirillum lipoferum* 1842	−	−	+	−	−
*Azotobacter vinelandii* (ATCC12837)	+	−	+	+	−
*Bacillus subtilis*	−	+	+	+	−
*Bacillus thuringiensis* subsp. *oloke*	−	−	+	−	−
*Bradyrhizobium japonicum* 3I1b6	−	−	+	−	−
*Enterobacter agglomerans* (ATCC23216)	+	−	+	−	−
*Enterobacter cloacae* CAL2	−	−	+	+	+
*Kluyvera ascorbata* SUD165	+	−	++	−	−
*Lactobacillus acidophilus* (ATCC 4356)	−	−	+	−	−
*Paenibacillus polymyxa* K56	−	−	+	+	−
*Pseudomonas brassicacearum* (ATCC 49054)	−	−	++	−	−
*Pseudomonas fluorescens* 34-13	+++	+++	++	−	+
*Pseudomonas protegens* CHAO	+	−	++	−	−
*Pseudomonas putida* (ATCC 12633)	+	−	++	−	−
*Sinorhizobium fredii* (ATCC51808)	+	−	++	+	−
*Sinorhizobium meliloti* RM11559	++	++	++	−	−
*Streptococcus salivarius* C699	−	−	+	−	−

## Data Availability

The data presented in this study are available on request from the corresponding author.

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
