# Peer review of "A Plant Biostimulant from Ascophyllum nodosum Potentiates Plant Growth Promotion and Stress Protection Activity of Pseudomonas protegens CHA0"

_plants, 2023, doi:10.3390/plants12061208_

Round 1

Reviewer 1 Report

This paper presents evidence that ANE (and components) regulates activity, gene expression and synthesis of essential products associated with root colonization in Pseudomonas protegens. Besides favoring root colonization, positive effects were also observed in growth promotion and salt mitigation in pea (Pisum sativum) plants. The study shows interesting data; however, experimental details were not fully provided and suggestions are made to improve result presentation and discussion. In general, the document should be checked by the authors and carefully edited.

Major changes:

1.     Results of screening concentration assays (0.5, 0.1 and 0.001% for ANE and 0.01, 0.05 and 0.001% for Fuc, Alg and Man) must be included as supplementary figures.

2.     No changes were observed in the amount of phosphate solubilized by P. protegens CHA0 in Alg treatment, please correct L135 indicating that alginate was found to inhibit phosphate solubilization.

3.     Authors mention that mannitol failed to enhance bacterial growth in salinity conditions L180-182. However, Figure 5a clearly shows a CFU increase. Please check congruency between image, graphic (Figure 5 a and b) and text (L180-181; L337-340). Figure 5 legend should also be also revised.

4.     Biofilms are involved in protecting bacteria in adverse environmental conditions, authors should also consider estimation of biofilm formation under salinity conditions. Without this information authors cannot establish a correlation between biofilm formation and root colonization, please rephrase (L 341-342).

5.     Fist line in the discussion section is incomplete (L265). 

6.     Authors should explain the lack of growth promotion in shoots in ANE and P. protegens (PP) under normal conditions (without salt stress) (Figure 7B).

7.     Authors explained the lack of a positive correlation between siderophore production and pvdS expression arguing that other genes might be involved (L300-302). What other genes? This should be discussed thoroughly.

8.     Authors should include a diagram displaying paths related to siderophore production, phosphate solubilization, HCN and IAA production by P. protegens CHA0, highlighting the up or down-regulation by ANE (or its components). If possible, also include the genes analyzed.

9.     Authors should difference between additive and synergistic effects through results and discussion sections.

10.   Essential information concerning hydroponics system (e.g. nutrient solution formulation and pH) and bacteria growing conditions are missing. Please complete sections 4.1 and 4.3.

11.   Include information regarding the number of Petri plates/samples analyzed per treatment during siderophore production, phosphate solubilization, HCN and IAA production assays. Examine description 4.7, what were fungal cultures used for? (L425). Please specify in all cases the number of bacteria (CFU) inoculated and the concentration of ANE, Fuc, Alg and added to culture media.

12.   How was ANE, Fuc, Alg and Man concentrations selected? Considering the proportion of these compounds in brown seaweeds or based in best results obtained in preliminary assays. Both options are mentioned in the manuscript (L434-436 and L120-122), please specify.

13.   Was nutrient solution supplemented with bacteria and other treatments (ANE, Fuc, Alg and Man) before salinity stress or at the same time? 

14.   Indicate the amount of total RNA used for cDNA synthesis.

15.   Include a paragraph describing Statistical Analysis.

Minor changes:

1.     P. protegens in all Figure legends and Pseudomonas in L311, L590, L622 should be italicized.

2.     Remove open parenthesis in L323.

3.     In Figure 8, please indicate at the vertical axis of each graphic the bacterial gene that is being analyzed.

4.     Include pea scientific name in section 4.1. Plant material and growth conditions (L377)

Reviewer 2 Report

The paper “A plant biostimulant from Ascophyllum nodosum potentiates plant growth promotion and stress protection activity of Pseudomonas protegens CHA0” reports an interesting study aimed at evaluating the effects of the extracts from the brown alga Ascophyllum nodosum, and of its main components (fucoidan, alginate and mannitol), on Pseudomonas protegens CHA0 secretion and colonization ability and the combined effects of the seaweed compounds and of P. protegens CHA0 on plant growth. The effects were tested on pea plants in the absence and presence of salt stress.

The topic is interesting, and the experiment was well conducted, however, I found several drawbacks and I think the manuscript in its present form is not suitable for publication.

-The characterization of bacterial strains which conducted to the selection of  Pseudomonas protegens CHA0 for the following analyses is well performed and described.

-The experiments on Pseudomonas protegens CHA0 treated with the different compounds, ANE, fucoidan, alginate and mannitol are interesting but when they are applied to plants, it is not clear how the root colonization was evaluated, in materials and methods it is not explained (paragraph 4.8)

In addition, regarding the hydroponic experiment, you should better describe the medium utilized for plant growth (paragraph 4.9). It is important to understand the amount of salt present in the medium, since also mannitol and alginate, which are in the form of sodium salt, could increase the amount of ions in the medium.

The phenotypes reported in Figure 6 are not well described and discussed, apart from the shoot fresh weight there are also: the number of plants effectively growing, the number of leaves, and the amount of chlorophyll, which could be considered.

Did you also check dry weight?

About mannitol, you mentioned its role as an osmolyte, and it can be synthetized also by plant cells for different purposes, including in response to salt stress. Please discuss in more detail the effects observed on pea plants when they were treated with mannitol also in combination with salt stress.

Lines 101-103 please rephrase, it is not clear

Line 220, ‘in absence of salt’ should be ‘in the absence of salt stress’  In the whole paper you should be more precise in this regard.

Line 221-223 please rewrite it is not clear. The expression levels are different, not the treatments.

Line 265 ‘affecting plant growth, health, and defense.’ I think this is an effect of cut and paste! Please correct.

Lines 321-324 please rewrite it is not clear. The open bracket is never closed.

Line 337 ‘1011’ should be 1011

Figure legends

Check that P. protegens is always in Italics

The panels are indicated with uppercase letters, while in the text they are reported with lowercase letters, chose one of them and be consistent.

Please explain the meaning of the different letters on each bar, in terms of statistical significance.

Figure 5-check the legend, there are some mistakes and ‘which dilution?’ is an internal comments? Please correct.

Figure 8 -You report the fold change values for each gene in the different conditions, and as described in the material and methods the 2-DDCt method was utilized.  Therefore, each control should have a value of 1, since it has been used to normalize the data, but in your graphs, it is not this way. Could you explain?

Statistical analysis

You chose to perform One-way Anova, but I think that you should have performed also two-way Anova, in order to evaluate more factors, especially for the data in Figure 7.

Table 2, please add the accession number of each gene tested, and the function.

The symbol of the housekeeping gene should be the same in the table as in the text, please check. Did you check the stability of the expression of this housekeeping gene?

Reviewer 3 Report

Microorganisms have a significant potential for obtaining effective and environmentally friendly plant growth regulators and stress-mitigating products based on them. Research aimed at improving these properties through synergy with other biological products is relevant today and arouses scientific and practical interest. And also there are some questions.

 Was it known about the beneficial properties of 17 bacteria before working with them?

Why was phosphorus solubilization, the production of IAA, siderophores and HCN chosen to assess the biological activity of strains, and not, for example, nitrogen fixation, synthesis of antibiotic substances, etc.?

In the work, not the production of indolyl-3-acetic acid was determined, but all indole compounds were (reaction with Salkovsky reagent).

How was protease production determined? There is no description of this method.

Numerical values should be given in table 1.

The CHA0 strain has long been a well-known strain with known properties. Screening does not look natural, but is perceived as "ostentatious".

line129: 57.66 psu - ? what are the units of siderophoric activity?

"Acadian" contains phytohormones in its composition?

Pseudomonades were cultivated using ANE as an additive to the nutrient medium. How did the presence of Fe and P in its composition affect the siderophoric and phosphate-mobilizing activity of bacteria?

Fig. 3. OD (424 nm) - what does this value show in the context of the parameter being determined?

What is the IAA value of 0.1% ANE?

Fig. 5 (which dilution?)?

How was sterility preserved when assessing the effect of the components on the colonization of the pea root by the pseudomonas strain? How you distinguished the studied strain from other bacteria?

It is recommended to transfer the genetic part higher to the biochemical one (it will be clear why the criteria for biochemical selection were chosen), and then experiments with plants.

Why was the criterion of siderophoric activity chosen? In your case, there is no competition at all for iron (or for other metal ions)?

Does the strain you are investigating produce other phytohormones other than IAA? And is the possible production of IAA confirmed at the gene level?

18: “after salt stress» or “under salt stress”?

53: Phaeophyceae — in italics.

132-136: The paragraph is not aligned in width.

143, 188, 189, etc.: P. Protegens - in italics.

Check the captions (a lot of typos) For example, "Alg, 0.01% alginate Man," is missing";" between "alginate" and "Man" .

Table 1: Which specific strains under which collection numbers are presented in the table (in particular, Azospirillum lipoferum, Azotobacter vinelandii, Bacillus subtilis)?

265: A new sentence begins with a capital letter.

Round 2

Reviewer 2 Report

The authors took into consideration all the suggestions, I think the manuscript can be published in its revised form

Author Response

Thank you for kindly review the MS and for helping us to improve it.